# Modelling *Drosophila suzukii* Adult Male Populations: A Physiologically Based Approach with Validation

**DOI:** 10.3390/insects11110751

**Published:** 2020-10-31

**Authors:** Luca Rossini, Mario Contarini, Federica Giarruzzo, Matteo Assennato, Stefano Speranza

**Affiliations:** Dipartimento di Scienze Agrarie e Forestali (DAFNE), Università degli Studi della Tuscia, Via San Camillo de Lellis snc, 01100 Viterbo, Italy; federicagiarruzzo@yahoo.it (F.G.); matteoassennato@yahoo.it (M.A.); speranza@unitus.it (S.S.)

**Keywords:** growth models, crop protection, integrated pest management, age-structured models, generalized Von Foerster’s equation, decision support systems

## Abstract

**Simple Summary:**

*Drosophila suzukii* is an injurious insect pest infesting soft fruit cultivations worldwide. In the recent years, it has spread in new territories, favored by its capability of adapting in new environments. Because of the severe reduction of the yields in cultivated fields, mainly due to the feeding and ovipositional activities, several studies were conducted on the side of the control strategies. However, the cost of the active ingredients, and the safeguard of the environmental and human health led scientists to introduce new tools for decision making. Among the latter, mathematical models play an important role, since their capability to summarize a pool of biological information such that the relations between the species and the external environment. The development and validation of mathematical models which faithfully describe the life cycle of ectotherms, as insects are, provide reliable tools to use in an integrated pest management framework. This work extends the application of a physiologically based model in the case study of *D. suzukii*, providing a validation in the case of an insect pest relevant in agriculture, and an accurate estimation of a set of biological parameters.

**Abstract:**

The Spotted Wing Drosophila (SWD), *Drosophila suzukii* (Matsumura), is a harmful insect pest for soft fruit cultivations. Even though its main hosts belong to the genera *Prunus* and *Rubus*, its high polyphagy and adaptability to new environments makes it a serious problem for farmers worldwide, who have reported several economic losses because of this pest. A wide series of proposals to control SWD are available and operate in line with the mechanisms of integrated pest management, demonstrating their high efficiency when applied at the opportune moment. This work aims to apply and validate a physiologically based model which summarises all the available information about *D. suzukii* biology, such as the relationship between environmental temperature and its development, fertility and mortality rates. The model provided, as a result, a description of a population of SWD females taking into consideration the multiple generations that occurred during the year. Simulations were then compared with field data collected in a three-year survey in two experimental fields located in the Sabina Romana area (Lazio, Italy). More specifically, *D. suzukii* males were monitored with traps in fields cultivated with mixed varieties of cherries and they were selected because of their clearer identification in comparison to females. Results showed a high level of reliability of simulations in representing the field data, highlighting at the same time that there is no discrepancy in simulating *D. suzukii* females in order to represent male populations.

## 1. Introduction

*Drosophila suzukii* (Matsumura) (Diptera: Drosophilidae), also known as Spotted Wing Drosophila (SWD), is a damaging insect pest for soft fruit cultivations worldwide. Its harmfulness is amplified by its high polyphagy and by its capability of colonising and adapting in new territories. For this reason, in fact, it has spread from Japan, its land of origin [1], to Asia, North and South America [2,3] and Europe [4]. More specifically, the first European report of the insect was in 2008 in Spain [5] and Italy [6] and since its introduction, its spread throughout the European continent was rapid. In particular in Italy, the first outbreak was recorded in Trentino Alto Adige region [7], causing significant yield losses among soft fruit cultivations. In Trentino Alto Adige, losses were estimated to be around 25–35% [7] of the yield, while in North America damages have been higher, ranging from negligible to 80% [8] depending on the area.

During the last decade, the spread of *D. suzukii* in Italy continued from the north to the centre of the country [6], favoured by the presence of wild and cultivated host plants.

Although cherry, strawberry and blueberry plants seem to be preferred by *D. suzukii*, infestations were also reported on apples, grapes, peaches and other species belonging to the genera *Prunus* and *Rubus* [2,7,9]. The damages caused by SWD to host plants are mainly due to ovipositional and to larval development; larvae feed on the endocarp tissue, making the fruits unmarketable and representing an access point for secondary hosts responsible for fungal or bacterial diseases [6].

Significant reductions of yields, with subsequent economic losses [10], are further amplified by the capacity of the insect to reach up to 15 generations in a single year [5]. In fact, in favourable conditions SWD can find refuge on wild vegetation and then reinfest the cultivated fields [7].

In normal conditions, adult females lay an average of 300–400 eggs [9] on ripening fruits utilising a serrated ovipositor. Even if females lay eggs on random fruits, it is possible for several females to lay up to three eggs in a single fruit. Eggs hatch after an average of two to three days [5], and larvae usually develop inside the same host fruit. The total larval stage duration is estimated to be around 8–13 days, and it is followed by a short pupation inside the same fruit or in the soil. Hence, in an average of 15–18 days, eggs become adults [11,12] ready for coupling and ovipositional in a range of two days after emergence [7]. The ovipositional of a single female can continue up to seven days and, because of the short duration of the whole life cycle, it is common to report the overlap of multiple generations.

Adults are 2–3 mm long and present a sexual dimorphism [5]: males are clearly identified by two black spots on the wings [3], while females can be confused with other *Drosophila* species, requiring optical instruments and expert technicians for their identification.

Depending on the type of diet, males have a higher thermal threshold (30 °C) than females (28 °C) although the former are not fertile at high temperatures [12]. On the whole, SWD can survive in thermal conditions ranging from 5 to 30 °C, with the optimum temperature for development and reproduction around 25–28 °C [7,13]. Accordingly, in the Mediterranean area its activity is highest in spring and autumn, with a significant reduction of the populations during hot and dry summers [14].

Since *D. suzukii* is responsible for significant yield reductions, a series of control strategies have been proposed to limit outbreaks and reduce the damages caused. Most of these are based on the use of agrochemicals such as wide-spectrum insecticides [6,15], but the increasingly restrictive laws in terms of pesticide usage is motivating researchers to investigate alternative control strategies, such as the use of natural enemies [16,17,18,19,20].

For this purpose, to favour the activity of SWD natural enemies, a series of agronomical strategies can also be used, such as a correct management of the wild vegetation surrounding the cultivated fields [21], the use of less susceptible plant varieties or with an early or delayed maturation [22].

The aforementioned control strategies need careful planning which may be supported by mathematical models. Models describing *D. suzukii* populations were first proposed by Coop et al. [23] and Damus et al. [24], with the aim of using them in decision support systems for farmers and technicians.

This study aims to apply and validate the physiologically based model introduced by Rossini et al. [25,26] in the case of *D. suzukii*. In particular, this model describes populations of ectotherms which develop over time and through their life stages, including biological information and environmental factors. Hence, on the basis of these features, it can be a reliable decision support system candidate which can incorporate the already available information about fertility and mortality rates [15] or the relationship between environmental temperature and mean development time [12]. The simulated populations were then validated with data collected in a three-year survey conducted in two cultivated cherry fields located in the Sabina Romana area (Lazio, Italy). The model response was focused on a restricted period of the year (April–July, depending on the season) when the fruits (in the fields surveyed) develop, become susceptible to SWD attacks and require the most control strategies.

## 2. Materials and Methods

### 2.1. Males Population Modelling

The application of a mathematical model for decision support systems in agriculture usually requires a simplification of the real phenomenon. Thus, before introducing the model, it is important to point out the main compromise of this work: the choice to simulate only male populations. The reasons behind this choice are supported by two relevant biological features of the SWD. As already stated, males are clearly identifiable by black spots on the wings, while females require a deeper analysis of the ovipositor in order to be distinguished from the other species of *Drosophila*. Accordingly, it was simpler to educate technicians to collect the experimental data, without changing the meaning of the experimentation. On the other hand, Emiljanowicz et al. [27] reported as result of their study a sex ratio of 1:1 (males: females) for the SWD. Hence, with good approximation, it is possible to consider the male population as similar to the female one.

With this precondition, let us introduce the physiologically based model applied to describe *D. suzukii* populations developing over time t and through the life stages x. The population density function, hence, is represented by a two-variables function N(t, x), solution of the following first-order partial differential equation [25,26,28]:(1)$$\{\begin{array}{c} \frac{\partial }{\partial t}N\left(t,x\right)+\frac{\partial }{\partial x}\left[G\left(t,x\right)N\left(t,x\right)\right]=-M\left(t,x\right)\;N\left(t,x\right)\\ N\left(t,0\right)=\overset{x_{m}}{\underset{0}{\int }}\beta \left(t,x\prime \right)\;N\left(t,x\prime \right)\;dx\prime \\ N\left(0,x\right)=n^{0}\left(x\right) \end{array}$$

Equation (1) considers all the fundamental elements for a reliable description of ectotherm populations. Firstly, the development through the life stages is driven by a “generalised development rate function”, G(t,x), which expresses, generally speaking, the relationship between the species, environmental parameters and other biological factors such as aging. Proceeding by order, the other relevant function containing biological information is the “generalised mortality rate function”, M(t,x). This function can consider all the biotic and abiotic factors which can express mortality, due to its dependence on time and age variables. The boundary condition, instead, describes the mechanisms of reproduction, since it represents the number of new individuals entering in the first stage. More specifically, reproduction is regulated by a “generalised fertility rate function”, β(t,x), which also depends in this case on time and age. Hence, the boundary condition provides the integer of the new individuals produced by the population between the physiological ages 0 and $x_{m}$. The last component of the equation is the initial condition, which expresses the age-class distribution of the population at time zero.

The Equation (1), hence, has the capability of summarising all the information available from other biological studies as, for instance, fertility, mortality and relationship with the external environment. However, it has been formulated with the broader hypotheses possible for the functions G(t,x), M(t,x) and β(t,x), as suggested by the word “generalised”. Nevertheless, to date there is some missing biological information and a series of simplifications is required. The main issue is related to the environmental parameters: in fact, there are no analytical functions to represent the variation of the daily average temperature or relative humidity throughout the year [29,30,31]. Accordingly, there is no possibility of calculating analytical solutions for the Equation (1), and it makes necessary the use of numerical solutions, in order to increase the precision of the simulations. A numerical scheme was proposed by Rossini et al. [25] in their first formulation of the model, and it provides for a combined use of the upwind scheme and Euler’s finite difference method [32]. More specifically, this methodology consists of a backward approximation of the partial derivative on the age variable, and a forward approximation of the partial derivative on time variable, providing the following scheme:(2)$$\{\begin{array}{c} N_{h}^{i+1}=\frac{i}{h}G_{h}^{i}N_{h-1}^{i}+N_{h}^{i}\left[1-2\frac{i}{h}G_{h}^{i}+\frac{i}{h}G_{h-1}^{i}-iM_{h}^{i}\right]\\ N_{0}^{i}=\sum_{h=0}^{h_{m}}\beta_{h}^{i}N_{h}^{i}h\\ N_{h}^{0}=N_{0} \end{array}$$

To avoid confusion between continuous and discrete variables, in the numerical scheme (2) the discrete age is represented by the h variable, while time is represented by i. The notation has also been changed for development, fertility and mortality rate functions, whose letters G, M and β were maintained, but time and age dependences are written as apexes and subscripts respectively.

The numerical scheme (2) is the discretization of the Equation (1) in its most general form but, as showed in the next sections, other simplifications will be required.

Since *D. suzukii* is a multivoltine species, the field monitoring of the first generation was inserted into the numerical scheme (2) as boundary condition, following the strategy already applied in the works of Rossini et al. [26,31,32]. Subsequently, the next generations will be simulated using the fertility rate function $\beta_{h}^{i}$, a discussion of which is explored in Section 2.2.

Numerical solutions can be calculated by inserting the numerical scheme (2) in any calculation software or implemented in a programming language code. Specifically for this work, the choice was to include the scheme (2) in a specialised software written combining C/C++ programming language with libraries from the open source software ROOT [33,34]. This calculation software was already introduced by Rossini et al. [26,29] and named *EntoSim*.

Analysing in greater depth the model (1) proposed to simulate SWD populations, it is possible to notice that it is more appropriate for describing a population of females, while the monitoring was conducted only on *D. suzukii* males because of their easier identification. Hence, as discussed at the beginning of the present section, it becomes fundamental to ensure that the model is able to represent multiple generations. Accordingly, even though the validation will be conducted comparing simulations with male populations, the simulations will be calculated considering a female population.

### 2.2. Development, Fertility and Mortality Rate Functions

#### 2.2.1. Development Rate Functions

Let us discuss in greater depth the development, fertility and mortality rate functions contained in the Equation (1) and its discrete scheme (2). Their role is fundamental for a correct application of the model because they contain the biological information about the species; however, as already stated in the Section 2.1, simplifications are required in this case.

In order of appearance, the generalised development rate function G(t,x) is the first expression to discuss. In its general form, it is able to express all the characteristics of the species depending on time and physiological age. In the literature, a series of expressions describing the relationship between the species and environmental temperature are available [35,36], but none of them include age variable dependence and other environmental parameters such as relative humidity or photoperiod. Despite this limitation, the approximation to consider temperature as main driving variable for the development of the insects through their life stages is reasonable, and has been used for a long time [35,36,37]. Hence, the G(t,x) function depends only on time, because temperature is assumed to be time-varying in an open field context.

By definition, the development rate is the inverse of the mean development time [37,38], and different authors have reported mathematical expressions to relate it to the external temperature. To pursue the aim of this work, the expressions that were considered were:The Brière development rate function [39]:
(3)$$G\left[T\left(t\right)\right]=aT\left(t\right)\left(T\left(t\right)-T_{L}\right){\left(T_{M}-T\left(t\right)\right)}^{\frac{1}{m}}$$
where T(t) is the environmental temperature, a and m are empirical parameters, $T_{L}$ and $T_{M}$ are the upper and lower thermal thresholds, respectively, below and above which the development of the species is theoretically not possible.The Logan development rate function [40]:(4)$$G\left[T\left(t\right)\right]=\psi \left[\exp\left(\rho T\left(t\right)\right)-\exp\left(\rho T_{M}-\frac{T_{M}-T\left(t\right)}{\Delta T}\right)\right]$$
where T(t) is the environmental temperature, ψ and ρ are empirical parameters, $T_{M}$ is the upper thermal threshold above which the development of the species is theoretically not possible, and Δt is the range between the maximum of the function and $T_{M}$.The Sharpe and De Michele development rate function [41,42]:(5)$$G\left[T\left(t\right)\right]=\frac{T\left(t\right)\cdot \exp\left(A-\frac{B}{T\left(t\right)}\right)}{1+\exp\left(C-\frac{D}{T\left(t\right)}\right)+\exp\left(E-\frac{F}{T\left(t\right)}\right)}$$
where T(t) is the environmental temperature, A, B, C, D, E, F are parameters related to the enzyme kinetics of the organism [25,43].

The estimation of the parameters of the Equations (3)–(5) requires a series of experiments in a climatic chamber at a constant temperature provided, in the case of *D. suzukii,* by Tochen et al. [12]. More specifically, data of this type are called “*life tables*”, and represent a powerful tool for knowing more about the biology of the insects [44]. The life tables published by Tochen et al. [12] were used to estimate the best fit function parameters through the *Minuit* algorithm contained in the ROOT software [45], as described by Rossini et al. [26,28].

Once the best fit parameters were available, the second step of the model application was the choice of the best representing expression for the life tables data through the “*a priori* analysis” proposed by Rossini et al. [26,28,30]. The method provides for the choice of the expression which reported the best result from the $\chi^{2}$ test and an $R^{2}$ value close to 1.

#### 2.2.2. Mortality Rate Function

Proceeding in order, let us discuss next the mortality rate function. A useful approximation of this expression was proposed by Asplen et al. [15], calculated on the basis of the survival rates from Tochen et al. [12], Dalton et al. [46] and Kinjio et al. [47]. In the Equation (1), the mortality rate function was presented in the generalised form M(t,x) including the dependence on both time and age variables. However, the function from Asplen et al. [15] only considers the mortality induced by the environmental temperature T(t), and, therefore, it does not consider the age variable. Mathematically speaking, mortality was described by the following second order polynomial convex function:(6)$$M\left(T\left(t\right)\right)=0.00035\cdot {\left(T\left(t\right)-15\right)}^{2}+0.01$$

#### 2.2.3. Fertility Rate Function

The remaining term of the Equation (1) to be discussed is the fertility rate β(t,x). A proposal of fertility rate function was provided by Asplen et al. [15], conveniently modified to be adapted to the Equation (1).

The first term of the fertility rate function is the Bieri expression [48]:(7)$$\beta_{1}\left(t_{ae}\right)=\frac{H\cdot t_{ae}}{L^{t_{ae}}}$$
where H and L are empirical parameters and $t_{ae}$ is the time after the adult emergence. By this hypothesis, time and age in the Equation (1) are different variables but, in light of the simplifications assumed for the development rate function, the time after adult emergence can be calculated as follows:(8)$$t_{ae}=t-\frac{1}{G\left(T\left(t\right)\right)}$$

In other words, $t_{ae}$ can be calculated by subtracting the egg-adult mean development time provided by the development rate function G(T(t)) from the time variable t. The Bieri function (7) describes the decrease in egg production rate of females as they get older, estimated at a constant temperature (21 °C in the case of *D. suzukii*).

However, the ovipositional rate is temperature-dependent, as showed by Tochen et al. [12]. Accordingly, there is need of a second term in the fertility rate function which considers the variations due to temperature, T(t), changes:(9)$$\beta_{2}=1-{\left[\frac{T\left(t\right)-T_{1}}{T_{2}}\right]}^{2}$$

In the Equation (9), $T_{1}$ and $T_{2}$ are empirical parameters, and it is a percentage reduction in fertility expressed by a concave function [15].

The third term involved in the fertility rate expression is the sex ratio, because in this way the portion of newly produced eggs, which will be able to provide the progeny of the next generation, is considered:(10)SR=0.5

Combining the expressions (7), (9) and (10), the fertility rate function applied to pursue the aim of this work was:(11)$$\beta \left(t,\;T\left(t\right)\right)=SR\cdot \frac{H\cdot t_{ae}}{L^{t_{ae}}}\cdot \left(1-{\left[\frac{T\left(t\right)-T_{1}}{T_{2}}\right]}^{2}\right)$$

The parameters H, L, $T_{1}$ and $T_{2}$ were retrieved by the work of Asplen et al. [15], and listed in Table 1.

### 2.3. Data Analysis and Comparison between Simulations and Field Data

The comparison was conducted using the $\chi^{2}$ function and the coefficient of determination $R^{2}$, since this methodology was successfully applied to provide a clear idea of the level of overlap between field data and simulations with the Equation (1) [25,26,28,30,32].

More specifically, the $\chi^{2}$ function is defined as:(12)$$\chi^{2}=\overset{n}{\underset{i=1}{\sum }}\frac{{\left(O_{i}-E_{i}\right)}^{2}}{E_{i}}$$
where $O_{i}$ is the i-th observed data, $E_{i}$ is the i-th simulated data and n is the total of the points to compare. The Equation (12) in this case assumes the meaning of indicator of the distance, according to the hypothesis that a higher $\chi^{2}$ means that there is less overlap between simulations and field data. On the other hand, it could be the case that with high populations even a small distance between two points can provide a high $\chi^{2}$ value, suggesting low reliability of the model, even if this is not verified.

For this reason, the coefficient of determination $R^{2}$ was introduced to support the $\chi^{2}$ value. Mathematically, it is defined as:(13)$$R^{2}=1-\frac{\sum_{i=1}^{n}{\left(O_{i}-E_{i}\right)}^{2}}{\sum_{i=1}^{n}{\left(O_{i}-{\bar{E}}_{i}\right)}^{2}}$$
where, again, $O_{i}$ is the i-th observed data, $E_{i}$ is the i-th simulated data and n is the total of the points to compare. An $R^{2}$ value close to 1 suggests that simulated data have a high probability of reporting the same trend of the field data, and for this reason it is a reliable indicator to flank to the function (12).

### 2.4. Experimental Design for Model Validation

The validation of the Equation (1) was conducted during the growing seasons 2017, 2018 and 2019 in two experimental fields located in the Sabina Romana area (Lazio, Italy), in the municipalities of Montelibretti and Monterotondo (only 2018 and 2019 seasons).

These fields were part of a wider monitoring net conducted by Agroambiente Lazio and were selected for the model validation because of the organic management of the plantation, and their closeness to the meteorological stations. The latter were managed by the ARSIAL agency (Regional Agency for the Development of Innovation and Agriculture in Lazio), which provided the temperatures in order to apply the numerical scheme (2). Moreover, since temperature data were provided on a daily basis, the smallest time step possible for the simulations was one day.

The experimental orchards had an average surface of 2000 square meters, cultivated with a mix of local cherry varieties, usually harvested in the months of May and June, and with similar agronomic management. The distance between the two orchards is approximately of 10 km.

In the centre of the aforementioned fields, three Droso-Trap (Biobest, Westerloo, Belgium) lured with Droskidrink (Azienda Agricola Prantil, Priò, Trento, Italy) were deployed from 20 April to 28 June in 2017, from 19 April to 12 July in 2018 and from 12 April to 18 July in 2019.

Traps were hung at a height of 1.5–1.8 m, and checked weekly, when, during each inspection, traps were emptied to count the number of adult males and the lure was renewed. According to the specifics provided by the constructor, three traps in each field were able to cover the whole surface. Hence, the field male population was represented by the average of the three traps.

Each weekly inspection provided only the number of SWD males, while the remaining part of the caught insects were roughly and randomly analysed only to ascertain the presence of *D. suzukii* females.

## 3. Results

Following the logic line described in Section 2, the first part of the work is concerned with the estimation of the parameters of the development rate functions using the life tables of SWD females reared on cherries reported by Tochen et al. [12]. Best fit parameters, $\chi^{2}$ and $R^{2}$ values are reported in Table 2, while the best fit functions are plotted in Figure 1.

Before proceeding to the simulations, it was necessary to individuate the most reliable development rate function among the aforementioned results, in order to use it in the numerical scheme (2). According to the *a priori* analysis, the Briére expression was the best-fitting function for the life tables, with a $R^{2}$ value close to 1 and P=0.99999 from the $\chi^{2}$ test. Other helpful information gathered from expression (3) were the optimal temperature for the SWD development, (28±1) °C, and the lower and upper thermal thresholds $T_{L}$ and $T_{M}$, respectively (Table 2). In order of reliability, the Logan (4) was the second-best expression, with a $R^{2}$ value close to 1 and P=0.99912 from the $\chi^{2}$ test. The coordinates of the maximum were also calculated for the Logan expression, providing the optimal temperature for the SWD development, (27±4) °C. This value is in accordance with the results from the Briére.

The Sharpe and De Michele (5) was the least reliable function in fitting the life tables, with a lower $R^{2}$-value (even though it was close enough to 1), and with P=0.98655 from $\chi^{2}$ test. The optimal temperature for the SWD development provided by the Sharpe and De Michele, (28±1) °C, is also in accordance with the Logan and Briére expressions.

Since the Briére resulted the best fitting function for the life tables, it was inserted into the numerical scheme (2) in order to evaluate the daily average temperatures provided by the meteorological stations.

The second step of the work was the simulation of *D. suzukii* populations in the two experimental fields. Since there was no monitoring available in the growing season 2017 for the Monterotondo field, it has not been simulated. Simulations of the Montelibretti field are reported in Figure 2. During this growing season, three generations were simulated, reporting peaks on 25 April, 14 May and 15 June. In comparison, in the experimental data, peaks were reported on 27 April, 11 May and 15 June.

The numerical evaluation of the reliability of the simulations is reported in Table 3: analysing the $\chi^{2}$ and $R^{2}$ values, it is possible to assess that there is a good accordance between the simulations and field data, even if the experimental peaks are delayed by two days in the case of the first generation and anticipated to be early by three days for the second one.

The second year of survey, 2018, presented different scenarios in the experimental fields. Two generations were simulated in the Montelibretti field (Figure 3), with simulated peaks on 27 June and 11 July, while the experimental data reported the same on 28 June and 12 July.

The situation assessed in the Monterotondo field reported only one generation with a simulated peak on 5 July, confirmed by the experimental data.

The $R^{2}$ and $\chi^{2}$ values listed in Table 3 highlighted a good response by the model in representing the experimental population, since the former are close to 1 and the latter are relatively low, when compared to the abundance of SWD populations.

The last year of the survey, 2019, was the most inhomogeneous when compared to the others. In the period March–July, four generations were reported in the Montelibretti field (Figure 4), with peaks on 25 April, 11 May, 11 June and 28 June, respectively, while the experimental populations, instead, reported the same on 25 April, 9 May, 13 June and 27 June.

In the Monterotondo field (Figure 4), three generations were reported, with simulated peaks on 27 April, 12 June and 4 July, respectively, while experimental data reported the same on 25 April, 13 June and 4 July.

Even in this case, the capability of the simulations to catch the experimental points was confirmed by the $\chi^{2}$ and $R^{2}$ values listed in Table 3. In particular, the higher level of reliability was reported in the Montelibretti field.

## 4. Discussion

The results demonstrate the good response of the simulations in representing the experimental populations of *Drosophila suzukii* on mixed cherry varieties cultivated in Central Italy. Considering the $R^{2}$ listed in Table 3, all the values of the three-year survey were higher than 0.9 except one case, which was close to 0.8. Accordingly, analysing the information provided by $R^{2}$, it is possible to assess that the model (1) has a good reliability in describing the trend of the experimental data. This result is also supported by the $\chi^{2}$ values, even if some clarifications about this are necessary. A high $\chi^{2}$ value does not always imply a lower reliability of the simulation, because it also depends on the abundance of both experimental and simulated populations. Accordingly, even small differences among high values can provide a higher $\chi^{2}$ [28]. This was the case in the year 2019 in the Montelibretti field, where the highest population density among the three years of survey was observed, and the corresponding $R^{2}$ value indicates that the simulation trend was well described.

In the light of the results obtained in the present study, the Equation (1) can be a valuable candidate to include in a decision support system tool, even if, to date, it has been validated only in two fields in Central Italy. At the same time, it is possible to point out that field monitoring is still fundamental for driving the model effectively. In fact, one of the most common issues in modelling pest populations is defining the initial conditions, namely, when the eggs of the first yearly generation are laid. This problem was circumvented with a monitoring of the overwintering adults, which restart their activity in the spring. Hence, the value of the first adults caught by traps can operate both as a time zero for simulation, and as a rough estimation of a coefficient of normalization for each generation. However, it may be possible that the deviation between simulations and monitoring data occurred in April during the three years of survey can be due to an earlier activity of the females, in respect of males [49,50]. Hence, in this case a monitoring on SWD females could improve the reliability of the simulations, taking into account that this is an important moment in IPM [51]. On the other hand, the advantage of an easier identification of SWD males is lost, thus requiring more qualified personnel and an adequate instrumentation.

Other important biological information can be retrieved by the development rate functions (3)–(5). In fact, the *a priori* analysis elected the Briére as the best representation of the life tables from Tochen et al. [12], but the other expressions can also provide helpful advice for an efficient management of *D. suzukii* outbreaks. For instance, the knowledge of the optimum temperature for the development of the SWD can be helpful to individuate a cherry variety with a phase-shifted ripening with respect to the seasonal SWD population peak. Taking as an example the experimental fields considered in this work, it is possible to assess that early spring varieties should be less susceptible to *D. suzukii* infestations, because of the less favourable temperature conditions for the development of the species [12]. Moreover, the optimal temperature values were calculated accurately, improving the information data already discussed by Ioratti et al. [7].

Even though there are some discrepancies among the Logan and the Sharpe and De Michele rate functions in representing the life tables from Tochen et al. [12], the optimal temperature values calculated are in accordance each other, and with the same value calculated on the Briére. Furthermore, the standard errors associated with the development rate functions are relatively low for the Briére and Logan, but higher in the case of the Sharpe and De Michele. The reason for this discrepancy is related to the number of the function’s parameters to estimate in respect to the number of points in the life tables. Six parameters for the Sharpe and De Michele development rate function makes inevitable a nonlinear regression with only one degree of freedom, too low to consider this result reliable. Hence, even if the *a priori* analysis had elected the Sharpe and De Michele rate function as the best representation for the life tables points, then the number of degrees of freedom would have been too low to consider reliable this result, and to use it for simulations.

The modelling approach described in this study, moreover, can be helpful to drive alternative control strategies based, for example, on the release of the species’ natural enemies.

Wang et al. [17] evaluated the thermal performance of two potential parasitoids of *D. suzukii* pupae, *Pachycrepoideus vindemiae* (Rondani) (Hymenoptera: Pteromalidae) and *Trichopria drosophilae* (Perkins) (Hymenoptera: Diapriidae). Hence, the availability of SWD natural enemy life tables allows the application of the Equation (1) in the same way showed for *D. suzukii*. Accordingly, a double simulation of the insect pest and its parasitoid/predator can help farmers and technicians to improve the parasitic/predation activity.

## 5. Conclusions

In conclusion, a final consideration is opportune in the light of the results of this study. The assumptions for the applications of the Equation (1), namely the use of female population parameters to simulate male populations, may seem at first sight not opportune, but the results of this study show that in open field conditions there are no differences in simulating SWD females to represent males population. This result is important because the model application needs the early detection of *D. suzukii* adults to estimate the time zero for the ovipositional; pointing the attention only on males reduces the possibility of confusion with other *Drosophila* species.

Beyond the promising results of this study, a considerable effort is still required to improve the model’s reliability. For instance, the validation of the model should be extended in other areas with different environmental features: in fact, it could be possible that when conditions change, the sex ratio of 1:1 is not still valid [52]. In this case, a simple modification of the model is required to mathematically describe SWD populations under different sex proportions. Moreover, the model has been validated in fields managed under organic regime, hence the effects of conventional pest management practices were not considered.

Future works should focus on the development of a model describing the effect of pest management practices on SWD population density, the winter diapause, the relative humidity on SWD development, as well as the effect of aging or of the photoperiod. Hence, this is a productive starting point, from which further improvements can be made.

## Figures and Tables

**Figure 1 insects-11-00751-f001:**
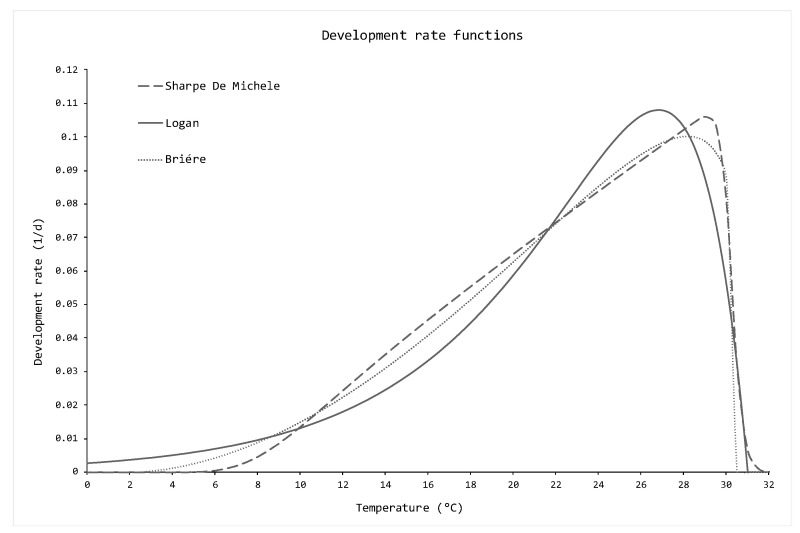
Graphical representation of Briére (3), Logan (4) and Sharpe and De Michele (5) best fit functions estimated with data from Tochen et al. [12]. Numerical values are available in Table 2.

**Figure 2 insects-11-00751-f002:**
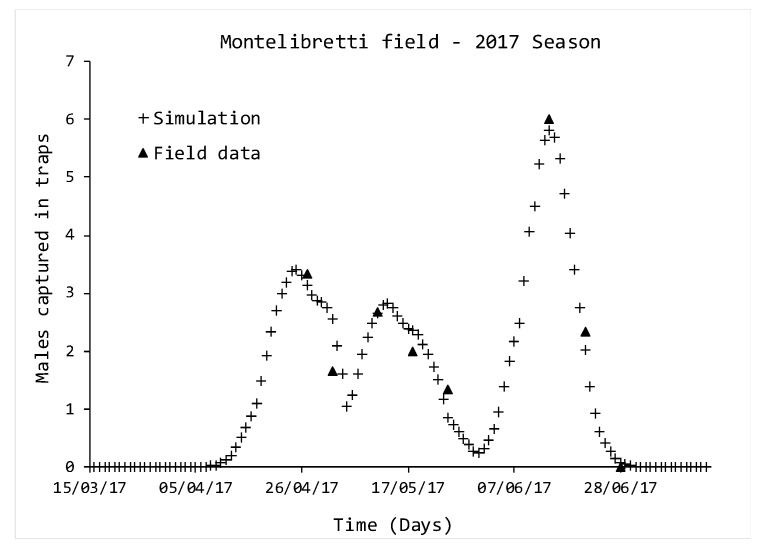
Comparison between simulated and monitored *D. suzukii* populations, in 2017. Triangle indicators represent data from traps, while crosses indicate simulations.

**Figure 3 insects-11-00751-f003:**
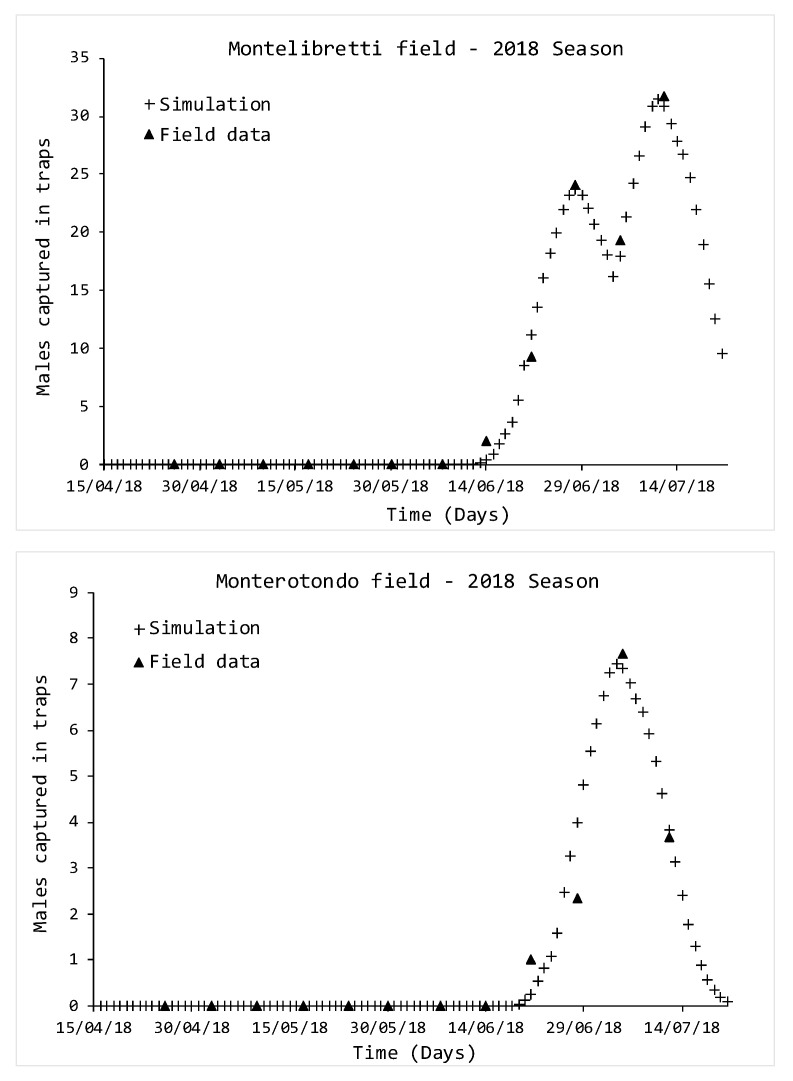
Comparison between simulated and monitored *D. suzukii* populations, growing season 2018. Triangle indicators represent data from traps, while crosses indicate simulations.

**Figure 4 insects-11-00751-f004:**
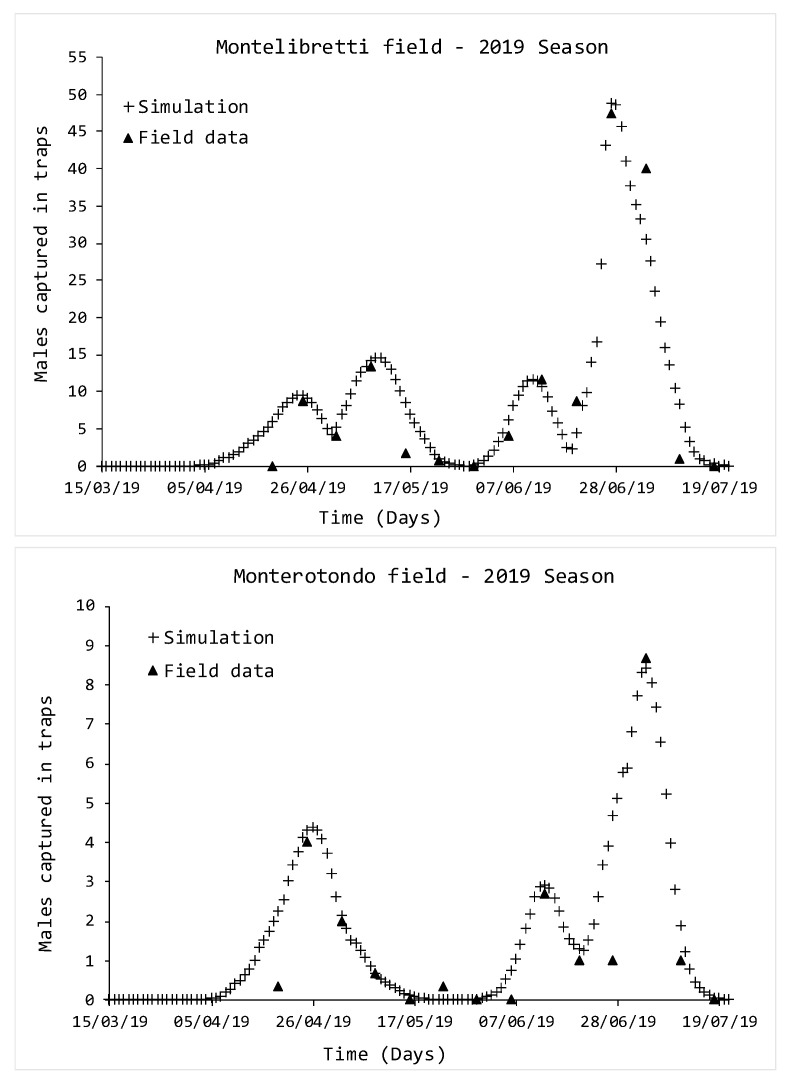
Comparison between simulated and monitored *D. suzukii* populations, growing season 2019. Triangle indicators represent data from traps, while crosses indicate simulations.

**Table 1 insects-11-00751-t001:** Fertility rate function parameters from Asplen et al. [15].

$\beta_{1}$		$\beta_{2}$	
Parameter	Numerical Value	Parameter	Numerical Value
H	0.585	$T_{1}$	20.875
L	1.0475	$T_{2}$	8.125

**Table 2 insects-11-00751-t002:** Best fit parameters (± standard error) of the Briére (3), Logan (4) and Sharpe and De Michele (5) development rate functions estimated with data from Tochen et al. [12]. For each nonlinear regression, the $\chi^{2}$ -value, the $R^{2}$ -value and the number of degrees of freedom (NDF) were reported.

Rate Function	Parameters	$\chi^{2}-\mathrm{Value}$	$R^{2}-\mathrm{Value}$	NDF (n)
Brière	$a=\left(1.20\pm 0.15\right)\times 10^{-4}$	0.001389	0.994	3
	$T_{L}=3\pm 2$			
	$T_{M}=30\pm 1$			
	m=6±3			
Logan	$\psi =\left(3\pm 1\right)\times 10^{-3}$	0.022202	0.886	3
	ρ=0.16±0.02			
	$T_{M}=31\pm 1$			
	ΔT=3±1			
Sharpe and De Michele	$A=\left(-1.2\pm 0.4\right)\times 10^{2}$	0.000284	0.946	1
	$B=\left(-3\pm 1\right)\times 10^{3}$			
	$C=\left(-1.2\pm 0.4\right)\times 10^{2}$			
	$D=\left(-3\pm 1\right)\times 10^{3}$			
	$E=\left(-1.1\pm 0.4\right)\times 10^{2}$			
	$F=\left(-3\pm 1\right)\times 10^{3}$			

**Table 3 insects-11-00751-t003:** Numerical consistence between simulations and field data of *D. suzukii* male populations.

Growing Season	Experimental Field	$\chi^{2}-\mathrm{Value}$	$R^{2}-\mathrm{Value}$
2017	Montelibretti	0.684	0.942
	Monterotondo	-	-
2018	Montelibretti	8.179	0.992
	Monterotondo	2.781	0.934
2019	Montelibretti	26.843	0.924
	Monterotondo	5.933	0.775

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
