# Peer review of "Modelling Drosophila suzukii Adult Male Populations: A Physiologically Based Approach with Validation"

_insects, 2020, doi:10.3390/insects11110751_

Round 1

Reviewer 1 Report

Rossini et al. described in this paper a theoretical model that could predict the population of SWD males collected in a trap in season. The manuscript is well written and bring interesting results for SWD population study. I have however some comments to improve the quality of the manuscript.

Line 268: information about practice in the orchard should be added (pesticides used, conventional, organic?). Did the orchards receive insecticides harmful for SWD in three years? Many more details should be provided to know how practice in the field could have influenced the population curve. Also, ideally, the distance between orchards should be added.

Line 278: authors said they monitored weekly the traps of SWD from March to July (5 months), could you explain why you have fewer field data (7 black triangles) in 2017-2018 than in the 2019 season.

Line 281: add renewal frequency of the lure.

Line 357: having two fields for validation is low. Usually, different fields in different regions with different pest management practiCe should be sampled to validate adequately predictions of a model. Could you discuss that and explain the limitation of your results. Also, rely on the pest management practice used in the orchards sampled to hypothesis on the fit of the model for other orchards.

Authors stated that the sex ratio is always 1:1 and that the population of males is similar to females, the literature showed that is not always true during the course of the summer season and I encourage authors to review the literature on SWD phenology on summer not only in Italy (the US and other European countries) to discuss their results. There is a limitation to the model use if the sex ratio is changing during the season and it will be important to state the limitation of using male population fluctuation for decision tool management

Author Response

Reviewer 1:

Revision overview:

English language and style:

I don't feel qualified to judge about the English language and style

Does the introduction provide sufficient background and include all relevant references?

Yes

Is the research design appropriate?

Can be improved

Are the methods adequately described?

Can be improved

Are the results clearly presented?

Yes

Are the conclusions supported by the results?

Must be improved

Comments and Suggestions for Authors:

Reviewer 1: Rossini et al. described in this paper a theoretical model that could predict the population of SWD males collected in a trap in season. The manuscript is well written and bring interesting results for SWD population study. I have however some comments to improve the quality of the manuscript.

Response: Dear Reviewer, we are grateful for your time dedicated to revise our manuscript, and for the helpful advises provided. During the revision, we carefully considered all the crucial points highlighted, with the hope to have reached an higher quality for our manuscript. We renew our availability for any further question or change required, thank you.

Reviewer 1: Line 268: information about practice in the orchard should be added (pesticides used, conventional, organic?). Did the orchards receive insecticides harmful for SWD in three years? Many more details should be provided to know how practice in the field could have influenced the population curve. Also, ideally, the distance between orchards should be added.

Response: Thank you for this comment. We have added the missing information in Section 2.1. The orchards (that have an approximate distance each other of 10 km) did not received any treatment during the three year survey. In fact, we decided to select only these two orchards because the agronomic practises were similar, as well as the local cherry varieties cultivated. Moreover, a third important element which determined the choice of these experimental fields was the proximity to the meteorological stations. Surely different agronomic practises can affect the population curves, and it is aim of future studies the consideration (with the addition of additional terms into the model) of the effects of pesticides on SWD populations.

Reviewer 1: Line 278: authors said they monitored weekly the traps of SWD from March to July (5 months), could you explain why you have fewer field data (7 black triangles) in 2017-2018 than in the 2019 season.

Response: Thank you for this comment. To avoid confusion we have reported the exact period, for each year, when traps were present in the field. Concerning the different number of points in the plots, there was an undesirable misprint that we have corrected during the revision. At the beginning, in fact, the idea was to restrict the plot ranges only when the population increases/decreases. This decision has been changed, since for a better comparison among the three years it was better to maintain the same time range for all the plots. We hope to have resolved this inconvenience, improving the readability of the results. Thank you again.

Reviewer 1: Line 281: add renewal frequency of the lure.

Response: Thank you for this comment. We have added the missing information in Section 2.1. The lure was renewed each week, in conjunction with the weekly inspection.

Reviewer 1: Line 357: having two fields for validation is low. Usually, different fields in different regions with different pest management practice should be sampled to validate adequately predictions of a model. Could you discuss that and explain the limitation of your results. Also, rely on the pest management practice used in the orchards sampled to hypothesis on the fit of the model for other orchards.

Response: Thank you for this comment. A validation in two experimental fields is not a high value, but at the same time we have selected the fields which have similar management and cultivated cherry varieties. Surely the comparison with other fields with different management can be a goal for future works, also to investigate how different agronomical practises and pest management can affect SWD populations. This work in this sense can be considered as a pilot study, where a series of helpful information are collected for entomologists and model scientists who aim to apply the model. Ongoing studies are focusing on other aspects to include in the model, such as the effect of different pest management. Thank you again for this suggestion.

Reviewer 1: Authors stated that the sex ratio is always 1:1 and that the population of males is similar to females, the literature showed that is not always true during the course of the summer season and I encourage authors to review the literature on SWD phenology on summer not only in Italy (the US and other European countries) to discuss their results. There is a limitation to the model use if the sex ratio is changing during the season and it will be important to state the limitation of using male population fluctuation for decision tool management.

Response: Thank you for this comment. Yes, it may happen that in some periods of the year, and in different climatic conditions the sex ratio can be different. For the Italian situation, for instance, an earlier activity of females in respect of males can happen in early spring, where the first generation of the year usually occurs. This could be a limitation for the model, since monitoring influence the initial conditions, but we also have to consider that the model should be a reasonable representation of a real phenomenon, whose basic assumptions are a compromise. Surely in different climatic condition it could be possible to have a different sex ratio, but this knowledge can be taken into account by the model, as well as a shift in males/females populations. For this reason, optimisation algorithms have to be object of future studies, because in this way we can exploit the field data to increase as much as possible the reliability of the model.

Reviewer 2 Report

The manuscript is well written and provides a practical model for D. suzukii that will aid in its field management.  Most of my comments are minor as in my opinion the model is well described and sound.

Minor comments:

The introduction is a bit wordy and can be condensed.

P6,L234: Change 'to' to 'from.'

P6, L328: The y-axis is 'Adult male population density', but the traps do not measure density. It would be more accurate to call those males found in traps 'males captured in traps.'

P10, L343:In both field locations there was a fairly substantial deviation between the model and the trap captures in early April that needs t be pointed out. One of the most powerful uses of a simulation model for D. suzukii will be to estimate the onset of activity so that management can be initiated. This may be a situation where female activity in the field may preceede that of males.

P11, L356: Need to provide the d.f. for the X² statistics, this also aids in their interpretation.

Author Response

Reviewer 2:

Revision overview:

English language and style:

Moderate English changes required

Does the introduction provide sufficient background and include all relevant references?

Yes

Is the research design appropriate?

Yes

Are the methods adequately described?

Yes

Are the results clearly presented?

Can be improved

Are the conclusions supported by the results?

Can be improved

Comments and Suggestions for Authors:

Reviewer 2: The manuscript is well written and provides a practical model for D. suzukii that will aid in its field management.  Most of my comments are minor as in my opinion the model is well described and sound.

Response: Dear Reviewer, we are grateful for the time dedicated to revise our manuscript, and for the helpful advises provided. We have carefully read all the comments and we made corrections accordingly, with the hope to have reached an higher quality for our manuscript. We renew our availability for any further question of correction needed. Thank you.

Minor comments:

Reviewer 2: The introduction is a bit wordy and can be condensed.

Response: Thank you for this comment. During the revision we have moved part of the information reported in the introduction in the discussion, with the aim to have improved the readability of the manuscript.

Reviewer 2: P6, L234: Change 'to' to 'from.'

Response: Thank you to highlight this misprint. We have corrected accordingly.

Reviewer 2: P6, L328: The y-axis is 'Adult male population density', but the traps do not measure density. It would be more accurate to call those males found in traps 'males captured in traps.'

Response: Thank you for this suggestion. This is a right observation which makes clearer the reading of the plots where results are reported. We have changed the y-axis names as suggested.

Reviewer 2: P10, L343:In both field locations there was a fairly substantial deviation between the model and the trap captures in early April that needs to be pointed out. One of the most powerful uses of a simulation model for D. suzukii will be to estimate the onset of activity so that management can be initiated. This may be a situation where female activity in the field may precede that of males.

Response: Thank you for this comment. This is a very interesting point that we have considered during the revision. In fact, the results highlight that more or less there is no difference in simulating SWD female populations to represent males, but in some critical periods, such as early April, a monitoring on females can provide a better estimation of the initial conditions needed by the model.

Reviewer 2: P11, L356: Need to provide the d.f. for the X² statistics, this also aids in their interpretation.

Response: Thank you for this suggestion. Surely the addition of the number of degrees of freedom may help the reader to interpretate the results, however in this case we have choose to not report the NDF because it is a bit out of context. In fact, in this case the X² is just an indicator of the distance between the simulated and field data points. Hence, the meaning of X² is not of a statistical test, but it is just an “overlapping index”. For this reason, the coefficient of determination R² was added. We have explained this difference in the text: the best representing development rate function for D. suzukii life tables was selected considering the coefficient of determination R² (we expect that a good fit has a value close to 1) and the X² function as statistical test. In this case, as shown in Table 2, the NDF was reported, as well as the corresponding P values in the results section. When we compare the simulation outputs with the field data we cannot use the X² in the same way as in the case of non linear regressions. There are several proposals to evaluate the “overlapping index” of a simulation, such as the coefficient of determination R², the root mean square error (RMSE), or the residual sum of square (RSS). The X² function, also, belongs to this family of expressions, going a bit outside of its “statistical meaning”, and if flanked with the R² can provide a numerical evaluation of the goodness of the simulations.
